

# Assessing the Impact of a Future Volcanic Eruption on Decadal Predictions

Sebastian Illing[1], Christopher Kadow[1], Holger Pohlmann[2], Claudia Timmreck[2]

[1]Freie Universität Berlin, Institute of Meteorology, Berlin, Germany
5   [2]Max Planck Institute for Meteorology, Hamburg, Germany

*Correspondence to*: Sebastian Illing (sebastian.illing@met.fu-berlin.de)

**Abstract.** The likelihood of a large volcanic eruption in the future provides the largest uncertainty concerning the evolution of the climate system on the time scale of a few years; but also an excellent opportunity to learn about the behavior of the climate system, and our models thereof. So the question emerges how predictable is the response of the climate system to future eruptions? By this we mean, to what extent will the volcanic perturbation affect decadal climate predictions and how does the pre-eruption climate state influence the impact of the volcanic signal on the predictions? To address these questions, we performed decadal forecasts with the MiKlip prediction system in the low-resolution configuration for the initialization years 2012 and 2014, which differ in the Pacific Decadal Oscillation (PDO) phase among other things. Each forecast contains an artificial Pinatubo-like eruption starting in June of the first prediction year. For the construction of the aerosol radiative forcing, we used the global aerosol model ECHAM5-HAM in a version adapted for volcanic eruptions. We investigate the response of different climate variables, including near-surface air temperature, precipitation, frost days, and sea ice area fraction. Our results show that the average global cooling response over four years of about 0.2K and the precipitation decrease of about 0.025mm/day, is relatively robust throughout the different experiments and seemingly independent of the initialization state. However, on a regional scale, we find substantial differences between the initializations. The cooling effect in the North Atlantic and Europe lasts longer and the Arctic sea ice increase is stronger than in the simulations initialized in 2014. In contrast, the forecast initialized with a negative PDO shows a prolonged cooling in the North Pacific basin.



## 1 Introduction

More and more attention is paid to the research field of decadal climate prediction in the last decade (Meehl et al., 2009; Smith et al., 2007). This field tries to fill the gap between short (weather to seasonal) predictions on the one hand, and long-term climate projections on the other hand. A number of studies revealed that there is at least potential prediction skill in near-surface air temperature, precipitation, and also three-dimensional variables like air temperature or geopotential height (Goddard et al., 2013; Kadow et al., 2016; Stolzenberger et al., 2016). Recently, some institutions like the MetOffice or the German project for decadal climate prediction MiKlip (Marotzke et al., 2016), started issuing decadal climate forecasts for near-surface air temperature on a regular - but still experimental - basis (metoffice.gov, 2017; Smith et al., 2013; Vamborg et al., 2017).

The skill of decadal predictions is usually evaluated using hindcast simulations (e.g. Doblas-Reyes et al., 2013; Kim et al., 2012) and it is assumed that external forcing is known over the whole simulation period. In a real decadal forecast, this assumption is invalid because rapid forcing changes - like volcanic eruptions - cannot be predicted in advance. Hence, strong tropical volcanic eruptions (SVEs) are arguably the largest source of uncertainty for this type of prediction. They increase the stratospheric aerosol load which leads to a reduction of global mean surface temperature due to the reduced incoming solar radiation. For instance, after the tropical Pinatubo eruption in 1991, a global peak cooling of about 0.4K (Thompson et al., 2009) was observed. Regionally, although warm anomalies are found after SVEs, e.g. the winter warming over Europe in the first two winters after the eruption (Kirchner et al., 1999; Robock and Mao, 1992). Apart from temperature, SVEs also have an impact on atmospheric composition, atmosphere, ocean dynamics and on the hydrological cycle (e.g. Robock, 2000; Timmreck, 2012). The last major volcanic eruptions followed a period of reduced global precipitation (Robock and Mao, 1992; Gu and Adler, 2011; Iles and Hegerl, 2014), and volcanoes can also modulate the African and Asian Monsoon systems (e.g. Liu et al. 2016). Sea ice in the Northern hemisphere is also affected by large volcanic eruptions and SVEs can cause up to a decade of increased Arctic sea ice extent (Ding et al., 2014; Gagné et al., 2017). Gagné et al (2017) for example demonstrated that the sea ice response to SVEs is dependent on pre-eruption temperature conditions with warmer pre-eruption climate leading to a stronger sea ice response. There is some evidence that SVEs have a positive impact on the Northern Atlantic Oscillation (NAO) in the first few winters following the eruption (e.g. Ortega et al., 2015; Swingedouw et al., 2017), but recent studies suggest that this signal might not be that robust (Bittner et al., 2016; Ménégoz et al., in press). It has also been suggested that the positive NAO response could be better interpreted in terms of a deficit of negative NAO circulations (Toohey et al., 2014).

There are only a few studies which focus on how volcanic forcing impacts on decadal climate predictions. Meehl et al. (2015) showed in a multi-model study that the Pinatubo eruption led to a reduction of decadal hindcast skill in the Pacific sea surface temperatures. Timmreck et al. (2016) could demonstrate that neglecting volcanic aerosol in decadal predictions significantly affects hindcast skill for near-surface air temperature and leads to a skill reduction in most regions up to prediction year 5. Bethke et al. (2017) explored how possible volcanic future eruptions impact on climate variability under



RCP4.5 and found that the consideration of volcanic forcing enhances climate variability on annual-to-decadal timescales. However not every volcanic eruption influences the climate in the same way. Zanchettin et. al (2013) showed in a case study of the Tambora eruption in 1815 that near-surface atmospheric and oceanic dynamics are significantly influenced by climate background conditions. Furthermore, hindcasts with the atmosphere only HADGEM1 model (Marshall et al., 2009) showed

that the climate anomalies in the first post-volcanic winter over Europe are strongly dependent on the stratospheric conditions in early winter.

Historically explosive tropical volcanic eruptions have a statistical recurrence frequency of about 50 to 100 years (Ammann and Naveau, 2003, Self et al, 2006). With the Pinatubo eruption almost 27 years ago and the recent ongoing unrest of the Mount Agung in Indonesia, the question arises what would happen if a large volcanic eruption occurs nowadays and how

dependent are the results from the start year and respectively the initial climate state? In this paper, we investigate the response of different climate variables, including near-surface air temperature (TAS), precipitation (PR), number of frost days (FD), and sea ice area fraction (SIC), to an artificial Pinatubo-like eruption happening in June of the first simulation year, for two initializations which differ in their initial state. To quantify the volcanic effect we compare our multi-yearly forecasts with the simulations without a volcanic eruption performed with the MiKlip prediction system (Pohlmann et al.,

2013).

In section 2 we describe the models used and our experimental setup, while the results of our analysis for different variables are presented in section 3. In section 4 we draw conclusions and discuss our results.



## 2 Model description and experimental setup

### 2.1. Model description

We perform our Pinatubo forecasts with the baseline1 version of the MiKlip prediction system (Marotzke et al., 2016; Pohlmann et al., 2013) which is based on the coupled Max-Planck-Institute-Earth System Model (MPI-ESM) (Giorgetta et

al., 2013; Jungclaus et al., 2013). The MPI-ESM is an Earth System model, with atmosphere, ocean, and dynamic vegetation components. We use the "low resolution" (LR) configuration of the baseline1 system, which has a resolution of T63 with 47 vertical levels in the atmosphere and an oceanic resolution of 1.5° with 40 vertical levels. The atmospheric component ECHAM6 (Stevens et al., 2013) is initialized with full-fields of temperature, vorticity, divergence, and sea level pressure from the ECMWF atmosphere reanalysis (Dee et al., 2011). The oceanic component MPI-OM (Jungclaus et al., 2013) is

initialized using anomaly fields from the ECMWF ocean reanalysis system 4 (Balmaseda et al., 2013) including temperature and salinity. For decadal forecasting, a stand-by model system for rapid model-based assessment of the decadal scale climate impact is needed, in case of any major volcanic eruption. However any modification to the climate model itself requires a re-tuning (Mauritsen et al., 2012), a new control run with constant forcing to make sure the model simulates a stable climate, and a new ensemble of historical runs as a reference for assessing skill enhancement through initialization (Goddard et al.,

2013; Illing et al. 2014).

Here, a two-step modeling approach is applied to consider the effect of large volcanic eruptions in the MiKlip decadal prediction system. In a first step, the formation of volcanic sulfate aerosol and its corresponding optical parameters (Aerosol optical depth (AOD), effective radius (Reff)) are calculated from the initial stratospheric $SO_2$ injection with a global stratospheric aerosol model. In a second step, AOD and Reff are used as monthly mean forcing in the decadal prediction

system.

For the construction of the aerosol radiative forcing, we use the global aerosol model ECHAM5–HAM model (Stier et al., 2005) in a version adapted for volcanic studies (Niemeier et al., 2009) which agrees very well with measurements of  AOD in the visible range and the effective particle radius after the Pinatubo eruption. We decided to simulate a future Pinatubo-like eruption, because the Pinatubo eruption is the best-observed eruption ever and the second strongest of the historical time

period (1850 till today). In addition, the likelihood of such eruption is in the order of 50 to 100 years (e.g. Self, 2006). To compile volcanic forcing fields, we inject 17 MT $SO_2$ over 3 hours at the geographical location of Pinatubo (15° N, 120° E) into the stratosphere (around 30 hPa). The simulated AOD field resembles not surprisingly the observed pattern after the Pinatubo eruption (Fig. 1a).

### 2.2 Experimental setup

Figure 1a shows the aerosol optical depth (AOD) simulated with the global aerosol model for a Pinatubo-like volcanic eruption. We use the simulated AOD as the forcing component for the decadal prediction system. For our experiment, we perform two decadal forecasts for ten years with ten ensemble members each, using a lagged-day method. One forecast was



initialized around December, 31st 2012 (Pinatubo-2012) and the other one around December, 31st 2014 (Pinatubo-2014). We chose these initialization years because they are relatively close together and therefore have similar greenhouse gas forcing, which can be considered close to present day conditions, but they also differ in important climatic conditions. In December 2012 the Pacific Decadal Oscillation (PDO) and the North Atlantic Oscillation (NAO) were in a negative phase,

whereas the Pinatubo-2014 experiment is initialized with a positive PDO and a positive NAO (Fig. 1b, c). Other important climate modes like the El Nino Southern Oscillation (ENSO), or the Atlantic Multidecadal Oscillation (AMO), are in both experiments in a similar state (not shown). PDO and NAO are both important drivers of internal climate variability. A negative PDO phase is associated with below average temperatures in Pacific Northwest, British Columbia, and Alaska and an above average Indian summer monsoon (e.g. Mantua and Hare, 2002). Whereas a positive NAO indicates colder and drier

Mediterranean regions, and warmer and wetter than average conditions in northern Europe and the eastern United States (e.g. Visbeck et al., 2001). The difference in both variables enables us to investigate the influence of initial climate conditions on the volcanic response of the model in a 'nowadays' setup. As reference datasets, we use the MiKlip baseline1 experiments initialized at the same start dates (b1-2012 and b1-2014) but without volcanic aerosol.

In order to quantify the effect of volcanic aerosols, we calculate the differences of the ensemble mean simulations of

Pinatubo-2012 - b1-2012 (exp-2012) and Pinatubo-2014 - b1-2014 (exp-2014). We also calculate the difference between exp-2012 and exp-2014 to quantify the impact of the different initial conditions. Statistical significance is determined by using a two-sided t-test (Wilks, 2011).



## 3 Results

### 3.1 Air temperature

Figure 2 shows a forecast for the four-year running mean near-surface air temperature (TAS) for different regions, like it would be issued by the MiKlip project (Vamborg et al., 2017), together with our Pinatubo experiments. The Pinatubo-like eruption leads to a significant decrease in global mean temperature of about 0.2K on average in the prediction years 1-4 in both experiments. The temperature difference gets smaller in later years, but is significant until prediction years 7-10 (Fig. 2a and b). Globally, there is no evident difference between the two experiments. The situation is different in the North Atlantic (NA) (Fig. 2c and d). In the first prediction years, the volcanic aerosol leads to a temperature decrease of about 0.3K in both experiments. However, in the 2012 experiments, the temperature difference decreases in prediction years 4-7 and completely vanishes from years 5-8. This adjustment is mainly because the b1-2012 experiment shows a negative trend in later prediction years and this negative temperature trend is not evident in the Pinatubo-2012 experiment. In contrast, Fig. 2d shows a constant temperature difference between b1-2014 and Pinatubo-2014 for the whole prediction period. For Europe (Fig. 2,e, f) both initialization dates show a significant surface cooling in the Pinatubo experiments in the periods 1-4 and 2-5, but the cooling is more pronounced and stays significant longer until years 4-7 in the 2014 model runs. The difference between the two initialization dates 2012 and 2014 is strongest in the northern hemispheric spring and fall (not shown). In 2012 there is no significant temperature decrease in spring and fall visible due to the Pinatubo-like eruption, whereas the Pinatubo-2014 simulation shows significant temperature drops up to 0.4K on average. In the North Pacific basin (Fig. 2,g, h), where the PDO index is calculated, both experiments show a temperature drop induced by the Pinatubo-like eruption in the first prediction years with the strongest values in prediction years 2-5. However, in the 2012 experiment, which is initialized with a negative PDO, the temperature difference stays nearly constant and significant over the whole prediction period, while in Pinatubo-2014 TAS starts recovering in prediction years 3-6.

This disparity is also visible in the global maps in Fig 3, which shows TAS for prediction years 1-4 and 7-10. For both initialization dates, the Pinatubo-like eruption leads to significant cooling over most parts of the Tropics, North America, and the North Atlantic (Fig 3 a, c) for prediction period 1-4. Generally, the cooling effect is strongest over the continents and reaches up to 1K over North America. We found the most substantial differences between the initialization dates in Europe, Siberia, and East Asia (Fig 3 e). Especially in Scandinavia the earlier initialized run shows a slightly positive effect, whereas the latter one shows a strong cooling. Thus the simulations started with the initial conditions of 2014 (e.g. positive PDO and NAO) react stronger to stratospheric aerosols released by the Pinatubo-like eruption in these regions. In contrast, exp-2012 shows a significantly stronger cooling over Alaska. The negative PDO phase results in less warm air being advected from the North Pacific into this region, a transport which is especially important if solar radiation is weakened (Wendler, 2012). In later simulation years, the cooling effect is less pronounced in both experiments (Fig 3 b, d, and f). In exp-2012 some parts of the tropics are still significantly cooler, and in the North Pacific, a significant negative horseshoe pattern is evident. This pattern is missing in the 2014 initialized runs, but there we have strong significant cooling over northern Canada, Arctic



Siberia, and the North Atlantic. This finding is in alignment with Fig. 2f where we saw that the cooling in the North Atlantic is persistent throughout the whole simulation period.

Figure 4 shows a cross-section of the zonal mean air temperature (TA) averaged over prediction years 1-4. In both experiments, the cooling we found at the surface continues in the troposphere and is strongest in the tropical troposphere at between 100 and 400 hPa. Exp-2014 shows a warming in the upper troposphere at 100 hPa in the northern polar region, whereas exp-2012 shows a slight cooling in this region. In the lower stratosphere, the Pinatubo-like eruption leads to a warming of up to 1.4 K in both experiments. This warming is due to the absorption of solar near-IR radiation from the increased sulfate aerosol formed after the eruption (e.g. Houghton et al., 1996; Stenchikov et al., 1998).

## 3.2 Sea Ice

Gagne et al. (2017) recently showed that a decade of increased Arctic sea ice followed the last three large volcanic eruptions. Figure 5 shows the differences of the ensemble mean forecasts of sea ice area fraction (SIC) for the prediction years 1-4 for the sea ice maximum in March (top row) and the sea ice minimum in September (bottom row). Overall we see increased maximum values of SIC due to the volcanic eruption in both experiments, but the two initialization times differ in the affected local areas. On the one hand, exp-2012 shows increased values of SIC in the Bering Sea of up to 10% where there is no evident signal in exp-2014. On the other hand, the 2014 initialized experiment shows significantly increased SIC values in the Nordic Sea where the sea ice area fraction of exp-2012 is only slightly higher. This different behavior is not only evident in the first four prediction years. Figure 6a) to d) show the four-year running mean forecast for maximum SIC in the Nordic Sea area (30°E-90°E, 70°N-85°N) and in the Bering Sea (165°E-195°E, 55°N-70°N). The experiment initialized in 2014 reacts stronger to the Pinatubo like eruption in the Nordic Sea and has significantly increased maximum SIC values over the whole prediction period. On the other hand, the 2014 initialization shows nearly no response to the volcanic aerosol in the Bering Sea, whereas the Pinatubo-2012 experiment has increased SIC over the whole forecast period with significant values up to prediction years 5-8. The stronger response in the Bering Sea in the 2012 experiment could be explained through the negative values of the PDO index bringing colder temperatures to Alaska (Overland et al., 2012; Wendler et al., 2013) and this cooling is even more pronounced in the simulation with a Pinatubo-like eruption (Fig. 3). We also see strong differences between our two experiments in the four-year mean of SIC minimum. Exp-2012 shows no positive response of SIC to the volcanic eruption in the Arctic region (0°E-360°E, 70°N-90°N, Fig. 6e) and even a slightly negative tendency in local areas (Fig. 5a) in the first four prediction years. On the contrary, we see a significant positive response of maximum SIC in the 2014 initialized experiment in the whole Arctic which locally reaches values of over 10% in prediction years 1-4 (Fig. 5b). This signal decreases slowly and is significant until prediction year 4-7 (Fig. 6f). Screen and Francis (2016) stated that wintertime Arctic warming and sea-ice loss is larger during negative PDO phases, which could partly cancel out the cooling effect through the increased aerosol load in the 2012 experiment.

Gagne et al. (2017) stated in their recent study that the sea ice response is dependent on pre-eruption temperature conditions and that warmer pre-eruption climate leads to a stronger sea ice increase. The results shown in Fig. 6 do not corroborate their



findings. In fact, they show a slightly contrary tendency and regions with higher initial sea ice and lower temperature conditions (not shown) react stronger to the Pinatubo-like eruption. This could be a model dependent effect or a sampling effect due to the focus of only two initialization times in our study.

It is notable that there is a decreasing trend in all our simulations in the three regions, and that the trend is not affected by the Pinatubo-like eruption. That is, if there are increased values of SIC in one experiment, the difference stays nearly constant for all prediction years.

### 3.3 Frost days

Not only mean temperature values are influenced by volcanic aerosol, but also temperature extremes. Figure 7 shows the anomaly of frost days in our simulations for the prediction years 1-4, calculated after the recommendations of the Expert Team of Climate Change Indices (ETCCDI, Karl et al. (1999)). Through the increased AOD the number of frost days rises, especially over land, in the northern hemisphere in most regions. The spatial distribution and magnitude differ between the two initialization times. In exp-2012, the highest significant values are in Bering Sea, Eastern North America, the Nordic Sea, and over China. Whereas in the Pinatubo-2014 experiment, the frost days increase most over the whole of North America, Scandinavia, Nordic Sea, and East Asia. In general, exp-2014 shows a stronger reaction to the volcanic eruption except for the Bering Sea. The spatial distribution of both experiments is in a good agreement with the pattern of TAS. The total number of frost days stays enhanced over the whole forecast period and is much higher in the northern hemisphere compared to the southern hemisphere (not shown).

### 3.4 Precipitation

A critical aspect is the understanding of the volcanic impact on the hydrological cycle. It has been demonstrated that volcanoes modulate the African and Asian Monsoon systems (Liu et al., 2016; Oman et al., 2006), impacting on areas which are now home to ~60% of the world population. We see a clear reduction in global mean precipitation in both experiments (Fig 8 a, b). In the first four prediction years, the magnitude of the reduction is about 0.025mm/day. This behavior is in agreement with previous studies which examined historical volcanic eruptions (Gu and Adler, 2011; Iles et al., 2013; Robock and Mao, 1992). The effect decreases with prediction time, but stays significant for all lead-times. In each of the experiments, the drying effect is stronger over land than over the ocean (Fig 8 c-f). Precipitation decreases over land with about 0.04mm/day in the first four prediction years of the 2014 experiment. This is about twice as strong as the maximal decrease over the ocean, but recovers faster to non-eruption values. The precipitation reduction over land is only significant until prediction years 2-5, whereas the decline over the ocean remains significant until years 5-8 in the 2014 experiment or even over the whole simulation period as in the 2012 experiment. While the reduction of land precipitation is a direct feedback to the increased AOD, the precipitation changes over the ocean area temperature feedback (Iles et al., 2013). Similar behavior has been found in CMIP5 model simulations, but it turned out that the precipitation changes are


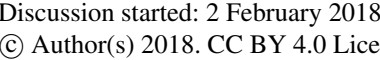


underestimated compared to observational data (Iles et al., 2013; Iles and Hegerl, 2014; Paik and Min, 2017). The latter suggest that this underestimation is connected to the underestimated latent heat flux in climate models.

Hence, while there could be some confidence in the general behavior of the post-volcanic changes in the hydrological cycle, the quantitative values of our forecast simulation should be taken with caution. Although the longer persisting reduction over

5  the ocean is seen in CMIP5 models, it cannot be detected in observations due to the short satellite time period, which covers only two major eruptions (Iles and Hegerl, 2014). The timescale of the precipitation reduction over the ocean is consistent with the response of TAS (Fig. 2). This is in agreement with previous studies (Iles et al., 2013; Joseph and Zeng, 2011). In the global precipitation-maps, we see a reduction of precipitation for both experiments through the volcanic aerosol in most regions in the first four prediction years (Fig. 9). The drying effect is strongest over the tropics, particularly in Southeast

10  Asia, and is even more pronounced in exp-2014. In general, exp-2014 shows a stronger drying response in the tropical region. In contrast, in this experiment, wetter conditions over Western Europe can be found which does not occur in exp-2012.





## 4 Summary and Discussion

In this study, we examined the sensitivity of decadal climate predictions to a tropical volcanic eruption using an artificial Pinatubo-like eruption as stratospheric forcing. Therefore we performed two decadal forecasts with different initial conditions, each forecast containing a Pinatubo-like eruption starting in June of the first prediction year, and compared them

to the corresponding simulations without a volcanic eruption. We chose the initialization years 2012 and 2014, because they differ in important climate indices like the NAO and the PDO. Other important climate modes like the El Nino Southern Oscillation (ENSO) or the Atlantic Multidecadal Oscillation (AMO), which have the potential to influence the volcanic response as well (e.g. Swingedouw et al. (2017) and references therein), are in both experiments in a similar state (not shown). We have shown that the global near-surface air temperature and precipitation decrease as a response to the volcanic

eruption is independent of the initial conditions and that the reduction is significant for the whole prediction period in both forecasts. In our experiments, the global mean temperature reduction in the first four years following a Pinatubo-like eruption is about 0.2K and the precipitation is about 0.025mm/day. In alignment with previous studies (e.g. Iles and Hegerl, 2014; Paik and Min, 2017) the drying effect is stronger over land than over the ocean, but the drying over land is only significant until prediction years 2-5.

Pre-eruption climate conditions play an important role for regional decadal predictions. We found significant regional differences between the two initialization experiments in the variables near surface air temperature, sea ice area fraction, frost days, and precipitation for the whole forecast period. One of the most substantial differences between the experiments can be found in the predictions of minimum and maximum sea ice area fraction. The volcanic eruption in the 2012 initialized simulation has nearly no effect on the four-yearly minimum SIC, whereas in exp-2014 we see a significant increase of up to

4%. For maximum SIC, both simulations show increased values, but the increase is concentrated in different regions (2012: the Bering Sea, 2014: Nordic Sea). This can be explained partly by the different phase of the PDO, as a negative PDO - as in the 2012 initialized experiments - brings colder temperatures to Alaska (Wendler et al., 2013) and strengthens the Arctic wintertime warming (Screen and Francis, 2016). In the 2012 experiment the temperature decrease in the North Pacific basin is nearly constant over the whole prediction period, whereas in 2014 the temperature starts recovering after a few years.

Additionally, we see a stronger cooling over Europe and a more pronounced drying in the monsoon region in the first four prediction years and longer lasting cooling effect in the North Atlantic in the 2014 initialized simulations. We also see a stronger increase in the number of frost days in most regions - except for the Bering Sea - in this experiment. We could not find a clear link between the different initial states of the NAO and one of these changes.

We note a few caveats and possibilities of improvement of this study. Recent model studies (Maher et al., 2015; Pausata et

al., 2015; Khodri et al., 2017) revealed that volcanic eruptions have a significant impact on ENSO and it would be interesting to see how decadal predictions are affected by this. Therefore our simulations in this study should be extended with experiments starting with other initial conditions like the recent El Nino year 2015/2016. Another factor currently neglected is the phase of the QBO as it changes due to the post volcanic atmospheric response (e.g. Thomas et al., 2009) and will be

modulated itself by strong volcanic eruptions (Aquila et al., 2014). The model (MPI-ESM) in the low-resolution version used in this study is not able to develop its own quasi-biennial-oscillation QBO, but the same model with higher vertical resolution shows predictive skill of the QBO of up to four years (Pohlmann et al., 2013). Another aspect is that our results could be model dependent and the analysis should be expanded to multi-model study. In order to gain a better understanding

of the impact of volcanic eruptions on decadal predictions and predictability, a collaboration is planned between the model intercomparison project on the climatic response to volcanic forcing VolMIP (Zanchettin et al., 2016) and the decadal climate prediction project DCPP (Boer et al., 2016). In line with the protocol of the upcoming CMIP6 (Eyring et al., 2016), a set of decadal prediction experiments will be conducted, where similar to our experiment, the impact of a Pinatubo-like eruption occurring in 2015 will be examined, which provides the unique opportunity to discuss our results in a multi-model

framework.

**Code and data availability**

The model output from all simulations described in this paper will be distributed through the World Data Climate Center https://www.dkrz.de/up/systems/wdcc and will be freely accessible through this data portal after registration. The code used for our analysis is available in a github repository and can be accessed through the following url

https://github.com/illing2005/future-pinatubo

**Competing interests**

The authors declare that they have no conflict of interest.

**Acknowledgments**

We thank Wolfgang Müller and Bereket Berhane whose comments helped to improve this manuscript. The research was

supported by the German Federal Ministry for Education and Research through the "MiKlip" program (FKZ: 01LP1519B (SI, CK), 01LP1517B (CT), and 01LP1519A (HP). We acknowledge the use of the European Centre for Medium-Range Weather Forecasts reanalyses data for the initialization (ORAS4, ERA-40, and ERA-Interim). Computations were carried out at the German Climate Computing Centre (DKRZ). Supporting information that may be useful in reproducing the authors' work are available from the authors upon request (sebastian.illing@met.fu-berlin.de).



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





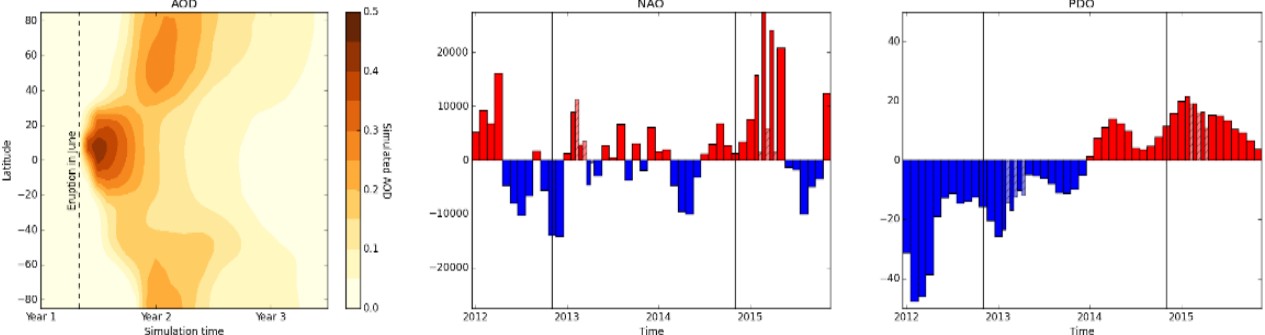

**Figure 1: a) Temporal evolution of the stratospheric aerosol optical depth (AOD) used for our simulations with a Pinatubo-like eruption. b), and c) show the climate indices NAO (a) and PDO (b) calculated from the assimilation model run. Hatched and lighter colors are the indices of the prediction system before the Pinatubo-like eruption. Vertical lines indicate the initialization time of our forecast experiments.**





**Figure 2: Time-series of four year running mean ensemble mean forecast of near surface air temperature anomalies (TAS). Anomalies are based on the period 1981-2010. The blue shows values without volcanic eruption, and red with a Pinatubo-like eruption. Left panel shows experiments initialized in 2012, and the right panel those initialized in 2014. Top row: global mean, second row: North Atlantic (60°W, 0°E, 50°N, 65°N), third row: Europe (10°W, 35°E, 30°N, 75°N), and bottom row: North Pacific Basin (130°E, 250°E, 20°N, 60°N). Dashed lines indicate significant differences between the values at the 5% level.**





Figure 3: **Differences of ensemble mean forecasts of TAS for prediction years 1-4 (left column) and 7-10 (right column). Top row shows exp-2012 (Pinatubo-2012 - b1-2012), middle row shows exp-2014 (Pinatubo-2014 - b1-2014), and bottom row shows the difference between two upper (exp-2012 - exp-2014). Crosses denote values significantly different from zero exceeding at a 5% level.**



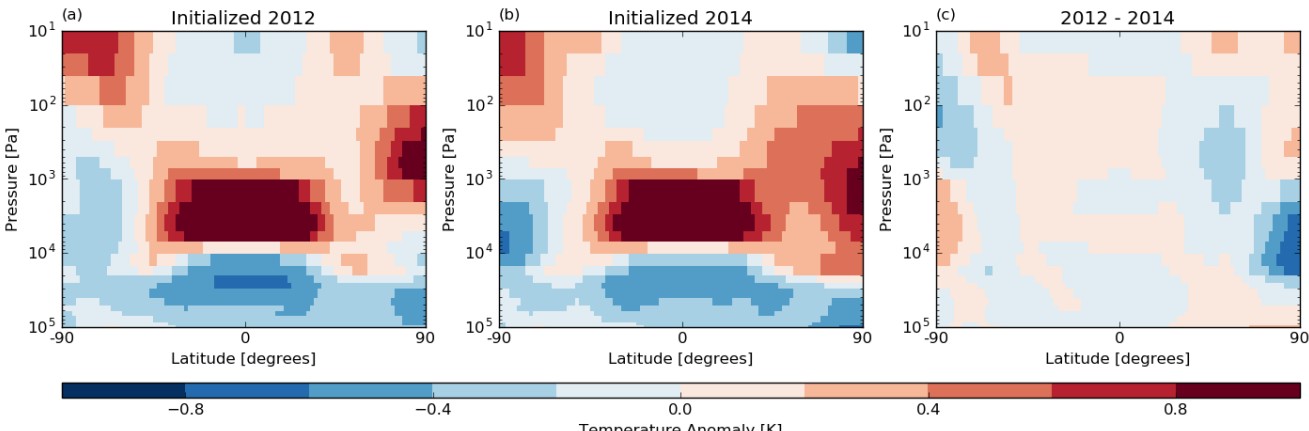

**Figure 4: Differences of ensemble mean forecasts of zonal mean air temperature (TA) for prediction years 1-4. a) exp-2012 (Pinatubo-2012 - b1-2012), b) shows exp-2014 (Pinatubo-2014 - b1-2014), and c) shows the difference between the two (exp-2012 - exp-2014).**





**Figure 5: Differences of ensemble mean forecasts of SIC for prediction years 1-4. Top row for the four year mean maximum in March and bottom row for the four year mean minimum in September. Left column shows exp-2012 (Pinatubo-2012 - b1-2012), middle column shows exp-2014 (Pinatubo-2014 - b1-2014), and the right column the difference between the two (exp-2012 - exp-2014). Crosses denote values significantly different from zero exceeding at a 5% level.**







**Figure 6: Same as Fig. 2, but for sea ice area fraction (SIC) maximum and minimum and other regions. Top row: Nordic Sea (30°E,90°E, 70°N, 85°N), middle row: Bering Sea (165°E,195°E,55°N,70°N), bottom row: Arctic (180°W,180°E,70°N,90°N). Dashed lines indicate significant differences between the values at the 5% level.**



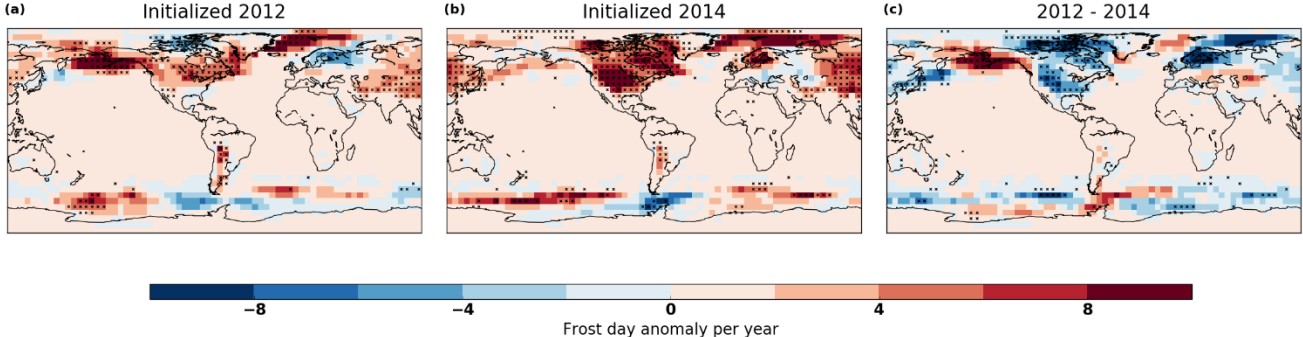

**Figure 7: Differences of ensemble mean forecasts of frost days (FD) for prediction years 1-4. Left shows exp-2012 (Pinatubo-2012 - b1-2012), middle shows exp-2014 (Pinatubo-2014 - b1-2014), and right the difference between two other (exp-2012 - exp-2014). Crosses denote values significantly different from zero exceeding at a 5% level.**

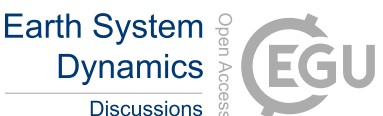



**Figure 8: Same as Fig. 2, but for precipitation (PR) and other regions. Top row: Global mean, middle row: ocean only, bottom row: land only. Dashed lines indicate significant differences between the values at the 5% level.**





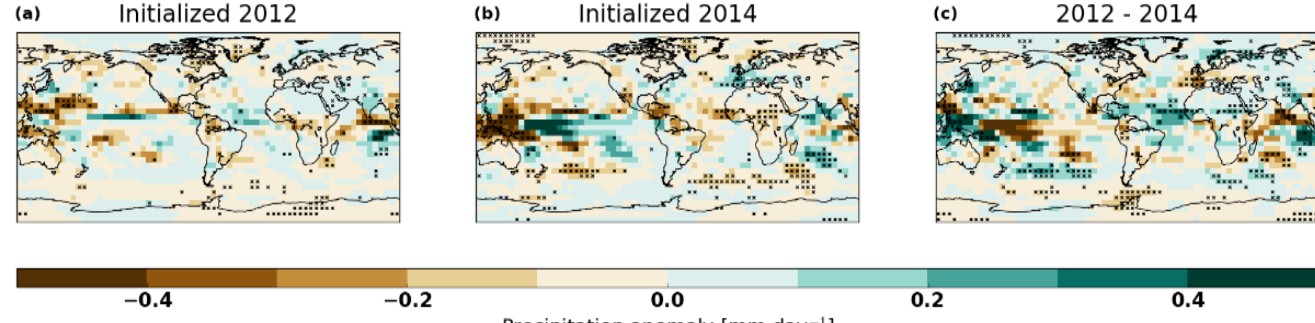

**Figure 9: Same as Fig. 7, but for precipitation anomalies (PR).**