# Peer review of "Assessing the Impact of a Future Volcanic Eruption on Decadal Predictions"

_Earth System Dynamics, 2018_

## Referee Comment (RC1) · Anonymous Referee #1 · 12 Mar 2018

The paper nicely explains the impact of volcanic eruptions on decadal predictions on a number of climatic variables. The authors find that although the global response (cooling and drying) to volcanic eruptions is independent of initial conditions, the regional response could be very different based on the initial state. I recommend acceptance with minor revisions as stated below.

Comment: An easy yet important addition to the paper would be to include the impact of the volcanic forcings on ENSO prediction. Since the data is already there it is just about computing some of the ENSO indices and presenting how they change due to volcanic forcings under different initializations.

[Figure]

2018.

---

## Referee Comment (RC2) · Anonymous Referee #2 · 12 Mar 2018

General comments This paper investigates how much effect a Pinatubo like eruption would have on decadal climate predictions for a number of variables, and whether the phase of major modes of variability (PDO, NAO) at the time of eruption make a difference to the climate response. The authors find that a volcanic eruption does indeed affect decadal predictions of temperature, precipitation and sea ice, and that some aspects are dependent on the initial state of the climate system, particularly at regional scales. This is an interesting and relevant topic, since the occurrence of a volcanic eruption would likely have a large impact on climate on the timescale of decadal predictions. The authors explain the background and relevance of the research well. I feel that the methods are appropriate to address the question and that the conclusions are supported by the evidence shown. The presentation of figures is generally good.

[Figure]

The article is within the scope of ESD and should be interesting to its readership. I feel that the article is worthy of publication, following the correction of some generally minor issues. My main suggested change relates to presenting ensemble spread to show how consistent results are between ensemble members, which will have implications for how predictable the climate response to eruptions is. Similarly, an indication of the extent to which modes of variability (particularly the NAO and PDO) remain in the same phase or diverge between ensemble members post eruption may shed more light on the differences between the two ensembles. My other suggestions mainly concern clarity in some places and English corrections.

Specific comments

Abstract - Mention that it is an ensemble of forecasts here.

Abstract, last line – which simulations have the negative PDO? The 2012 ones? Please make that clearer.

Abstract: "So the question emerges how predictable is the response of the climate system to future eruptions?" Looking at the ensemble spread would also help address this question- see also comments below. Whilst you may find a volcanic signal in the ensemble mean, the paper does not really give a feel for how consistent these signals are between ensemble members, and this would have implications for how predictable the climate response to volcanic eruptions is.

Introduction, first paragraph – is it worth mentioning briefly where the predictability comes from? E.g. is it from certain modes of variability? Or from SSTs? Etc. I assume this list of variables are the things that are able to be predicted rather than the predictors?

Pg 2 Line 20, also pg 8 line 20- and the south American monsoon e.g. Joseph and Zeng 2011??, Iles and Hegerl 2013;14

Pg 2 Line 21-22 – is this decade long sea ice response for the large eruptions that

occurred around the time of the Little Ice Age, or for twentieth century ones as well?

Pg 4 line 32 – What is a "lagged-day method" ? Can you explain briefly?

Pg 5 first paragraph and throughout – I found the naming of the two experiments some-what confusing (Pinatubo-2012 and Pinatubo-2014), because they are initialised on the last day of December in 2012 and 2014 respectively, meaning that the eruptions happen in 2013 and 2015 respectively, and that only one day of 2012/ 2014 is actually simulated. I wonder if they would be better off being called Pinatubo 2013 and 2015? Or at least make sure that it is clear that the eruptions happened in summer 2013 and 2015.

Figure 1a and pg 4 line 27 – is it possible to plot the observed equivalent of figure 1a if its not too much work?

Figure 1 b and c - Are the vertical lines showing the initialisation dates in the right place? They should be almost on the dashes for 2013 and 2015 on the x axis given the initialisation happens on the 31st December. - Could you indicate on the plots when the eruptions happen? - Could you explain explicitly why the hatched bars are different from the solid bars before the eruptions? – Is this because the initialised runs diverge from the assimilation run? How different is the phase of the NAO/ PDO between different ensemble members before the eruptions? Can this be shown on the plot? Also it would be interesting to see how the NAO and PDO indices evolve after the eruptions and how much this differs between ensemble members since this will impact the post volcanic climate response.

Page 6 line 2 "AS it would be issued by the Miklip project" – what do you mean? Is this how the Miklip forecasts are presented? Do you mean that they present things as 4 year means? If not, what is the rationale for presenting results as 4 year means when some aspects of the climate response to volcanoes e.g. the precipitation response over land, happen on a timescale less than 4 years?

Page 8 : Please define what a "frost day" is. Also, I am not sure whether it is really a climate extreme – i.e. in cold regions every day might be a frost day, and that would be normal, not extreme.

Page 9 line 8/9: " in most regions" I am not sure that a precipitation reduction can be seen in "most regions" in figure 9.

Figure2 - "Anomalies are based on the period 1981-2010" – is the climatology based on the assimilation run? - I assume these plots are showing annual means? - It could be interesting to show the ensemble spread as well. (Also for the other similar figures).

Figure 5: near-zero changes could be shown in white for clarity e.g. so it would be white where there is no sea ice all year round, or sea ice 100% of the time.

Figure 9 and related discussion (precipitation): -Is the difference in the Pacific related to the phase of ENSO? How does the phase of ENSO compare in the post eruption years between the two ensembles? (I know they started off in the same phase when they were initialised, but they may have deviated since) -Also, is the difference over Europe due to a different phase of the NAO? -I am also intrigued as to how similar or different the variability of modes like ENSO/ PDO/ NAO are within the members of each ensemble- i.e. the PDO and NAO start off in a similar phase, but how much do they diverge between ensemble members over time? -I also suspect that using a mean of the four post eruption years is not the most appropriate to look at land precipitation, because the response tends to last more like 2 years, unless presenting 4 year means is the standard for decadal predictions.

Pg 10 ENSO discussion: It would be worth mentioning that this influence of volcanoes on ENSO is debated.

Conclusions "We have shown that the global near-surface air temperature and precipitation decrease as a response to the volcanic eruption is independent of the initial conditions" – but only for the initial conditions that you have tested- this sentence currently sounds as if you mean its independent of all initial conditions, including things you haven't tested (e.g. ENSO phase), which you obviously can't comment on.

Technical corrections

Pg 1 line 9/10 "So the question emerges how predictable is the response of the climate system to future eruptions?" Add a colon between emerges and how. Same for page 3 line 9 -> "the question arises: What would happen..."

Line 12: "among other things" – if I understand correctly, the only other thing was the phase of the NAO, so this could be said specifically.

Line 21: "than in the simulations initialized in 2014" – delete "than", otherwise it means the opposite!

Pg2 line6 MetOffice -> Met Office , (assuming you mean the UK one)

Pg2 line 1 "more and more attention is BEING paid"

Pg 2 line 16 "regionally, HOWEVER, warm anomalies are ALSO found..." ("although" is not the right word and changes the meaning of the sentence). Is it worth saying why? i.e. its related to the positive NAO pattern that often occurs after eruptions.

Pg 2 Line 18 " atmosphere AND ocean dynamics"

Pg 2 Line 19 "The last major volcanic eruptions followed a period" -> "The last major volcanic eruptions WERE followed by a period" otherwise it means the opposite. Also, what do you mean by "the last major eruptions?" The ones that occurred this century? Same for pg 7 line 10

Pg 2 Line 25 – are these studies about the NAO observation based or model based?

Pg 2 Line 31 "could demonstrate" -> "demonstrated"

Pg 2 Line 34 "volcanic future eruptions" -> future volcanic eruptions

Pg 3 line 10 "from the start year" -> "on the start year"

Pg 3 line 13 "multi-yearly" -> "multi-year"

Pg 4 line 25 "in the order of 50 to 100 years" -> "in the order of ONCE EVERY 50 to 100 years"

Pg 5 line 7 "are in both experiments in a similar state" -> " are in a similar state in both experiments"

Pg 5 line 11 variables -> modes

Pg 5 line 12 "nowadays set up" sounds a bit strange. How about "present day set up" ?

Pg 7 line 8 "solar near-IR radiation BY the increased sulfate aerosol"

Pg 7 line 19 (and elsewhere) "reacts stronger" -> " reacts more strongly"

Pr 8 line 13 "Whereas in the Pinatubo-2014 experiment, the frost days increase most over the whole of North America, Scandinavia, Nordic Sea, and East Asia." I get the impression that this sentence should be joined to the previous one "…, whereas…" If not, the change "whereas" to "In contrast". "Whereas" needs both statements to be in one sentence. Also "Nordic Sea" -> "the Nordic Sea".

Pg 8 line 21 –" ∼60% of the world's population" This statistic needs a reference.

Pg 8 line 30 "area" -> "are a"

Pg 8 line 31 "but it turned out that the precipitation changes are…" sounds too colloquial. "but the precipitation changes were…" is sufficient.

Pg 10 line 2 – delete "Therefore"

Pg 10 line 8 "are in both experiments in a similar state" -> are in a similar state in both experiments AT THE TIME OF INITIALISATION. (see also comments for figure 9)

Pg 10 line 28 "We could not find a clear link between the different initial states of the NAO and one of these changes." "one of these changes" -> "any of these changes" or just "these changes".

Pg 11 line 8 "experiments will be conducted, where similar to our experiment, the impact" -> "experiments will be conducted where, similar to our experiment, the impact"

Figure 6 and 8 captions "other regions" -> " different regions"

---

## Referee Comment (RC3) · Anonymous Referee #3 · 17 Mar 2018

General Comments:

This manuscript describes a set of decadal prediction experiments initialized with different phases of the NAO & PDO where the effects of a volcanic eruption of the magnitude of Mt. Pinatubo are examined. The authors find expected responses in the temperature and precipitation fields in the global average, but find that the different initial states produce regional responses that are different. This is a nice result that has implications for how decadal predictions should be performed when a volcanic eruption happens. The manuscript can be improved in terms of clarity and improvements in figures.

The paper is more or less complete in terms of analysis except for one issue that they could have explored a little more. This pertains to the 2014-Pinatubo experiment where the years 1-4 precipitation response (Figure 9) indicates a pattern that is very similar

to an El Niño response. While this hints at a possible volcano triggered El Niño, they make no mention of it but discuss this possibility in the conclusions purely in other published references. It would be interesting to see if indeed the 2014-Pinatubo runs show that an ENSO event was triggered – while the 2012-Pinatubo doesn't.

Detailed comments:

1. Abstract. Line 12: A little more descriptive wording than just "the MiKlip prediction system" would be useful here.

2. Page 2, Line 1: Suggest changing "more attention is paid to the research field of decadal climate . . ." to "more attention is paid to decadal climate . . ."

3. Page 2, Line 18: The wording in "impact on atmospheric composition, atmosphere, ocean dynamics" does not make clear what other than composition is altered in the atmosphere.

4. Page 2, Line 24: Not clear what a "positive impact on the North Atlantic Oscillation" means. The following sentence does not state it any better when one reads ". . .the positive NAO response could be better interpreted in terms of a deficit of negative NAO circulations".

5. Page 3, Line 9: The use of "nowadays" sounds strange. Suggest rewording.

6. Page 3, Lines 9-10: The sentence ". . . how dependent are the results from the start year and respectively the initial climate state?" implies that initial climate state and start year are delinked.

7. Page 4, Line 3: The phrase "Pinatubo forecasts" should be changed or an uninformed reader may take it as a forecast of the eruption itself.

8. Page 4, Line 11: "For decadal forecasting, a stand-by model system for rapid model-based assessment . . .". Why the use of "stand-by"? Do you mean "operational"?

9. Page 4, Lines 24-25: Suggest changing ". . . of the historical time period (1850 till

today)" to "... since 1850".

10. Page 4, Line 32: "... using a lagged-day method" can be explained better in words or needs a reference.

11. Page 5, Line 1: Why "around"? Shouldn't it be "on"? How many days before and after Dec 31 are the other forecasts?

12. Page 5, Line 11: What does "both variables" refer to?

13. Page 5, Line 12: What is a "a 'nowadays' setup"?

14. Page 6, Lines 5, 6: The word "significant" here (at least in the first instance) needs to be changed to "statistically significant at x% level".

15. Page 6, Line 29: Does the phrase "in less warm air being advected ..." mean less advection or the air is less warm?

16. Page 7, Line 27: Suggest using "On the other hand" instead of "On the contrary".

17. Page 8, Line 5: Not sure the difference between what and what stays nearly constant?

18. Page 8, Line 24: Not clear what variable/difference "... stays significant for all lead-times."

19. Page 8, Line 31: Suggest rewording "Similar behavior has been found in CMIP5 model simulations, but it turned out that the precipitation ..." to "Similar behavior has been found in CMIP5 model simulations, although they underestimate the precipitation ...".

20. Page 9, Lines 9-10: "The drying effect is strongest over the tropics, particularly in Southeast Asia, and is even more pronounced in exp-2014. In general, exp-2014 shows a stronger drying response in the tropical region." I am not sure it is a drying all across the tropics. I see the pattern in exp-2014 as an eastward shift of precipitation

– a possible signature of El Niño. The interpretation of Fig. 9 presented needs to a relook and the authors can examine whether indeed this is an ENSO event in the model simulations.

21. Page 10, Line 8: The phrase "... are in both experiments in a similar state..." may be reworded as "are in a similar state in both experiments".

22. Page 10, Lines 31-32: "Therefore our simulations in this study should be extended with experiments starting with other initial conditions like the recent El Nino year 2015/2016." In your simulations, the ENSO response like pattern is seen for 2014-Pinatubo runs. The initial conditions for both runs were similar in terms of ENSO state - yet one of them produces what looks like an ENSO-like pattern. It should be relatively easy to check (and at least comment on) whether it triggered an El Nino. Initializing with ENSO conditions is not answering the question whether volcanoes might trigger ENSO events.

23. Page 24, Figure 7 caption: What does "two other" refer to?

Technical Comments:

24. Page 19, Fig 2 caption: The coordinates "North Atlantic (60°W, 0°E, 50°N, 65°N)" is better written as "North Atlantic (60°W-0°E, 50°N-65°N)". Similarly for other regions.

25. Figures 3,5, 7, 9 all have maps shown with no latitude/longitude markings or labels.

26. Figure 4: Vertical axes in hPa units would be better. Latitude axis can be shown with ticks/labels so effects in polar/tropical regions are better seen.

---

## Author Comment (AC1) · 19 Apr 2018

The paper nicely explains the impact of volcanic eruptions on decadal predictions on a number of climatic variables. The authors find that although the global response (cooling and drying) to volcanic eruptions is independent of initial conditions, the regional response could be very different based on the initial state. I recommend acceptance with minor revisions as stated below.

**Comment**: An easy yet important addition to the paper would be to include the impact of the volcanic forcings on ENSO prediction. Since the data is already there it is just about computing some of the ENSO indices and presenting how they change due to

[Figure]

volcanic forcings under different initializations.

We thank the reviewer for the constructive comment and useful suggestion. We added the suggested ENSO indices, in particular the temperature based NINO4 index and the ENSO precipitation index (ESPI), to the manuscript (Fig. 10). Indeed, we find some interesting behavior in these indices and extended section "3.4 Precipitation" with the following text:

"In the global precipitation maps, we see a decrease of precipitation for both experiments through the volcanic aerosol in large parts, especially over land, in the first four prediction years (Fig. 9). The drying effect is strongest over the tropics, particularly in Southeast Asia, and is even more pronounced in exp-2014. In fact, the tropical precipitation pattern in Southeast Asia and the East Pacific in exp-2014 is very similar to an El Nino response. Recent model studies (Maher et al., 2015; Pausata et al., 2015; Khodri et al., 2017) revealed that volcanic eruptions have a significant impact on ENSO and there is some ongoing debate whether tropical volcanic eruption can trigger an El Nino event (Meehl et al., 2015; Predybaylo et al., 2017; Swingedouw et al., 2017). To further investigate this, we calculated the temperature based Nino4 index (Trenberth and Stepaniak, 2001) and the ENSO precipitation index (ESPI, Curtis and Adler, 2000) for both experiments for the first four prediction years (Fig. 10) as twelve month running means to reduce variance. The ensemble initialized in 2014 with a Pinatubo-like eruption shows a tendency towards El Nino conditions, whereas the baseline1 ensemble favors a weak La Nina condition (Fig. 10 b, d). The difference between the two experiments in the ESPI is significant until simulation months 18-30 when both indices come back to neutral conditions. In exp-2012 there is no difference evident in the first three prediction years, but in year four the baseline1 ensemble starts simulating a La Nina phase (Fig. 10 a, c) with a significant difference to the Pinatubo-like experiment. In general, exp-2014 shows a stronger drying response in the tropical region. In contrast, in this experiment, wetter conditions over Western Europe can be found which does not occur in exp-2012 (Fig. 9)."
**References**

Curtis, S. and R. Adler, 2000: ENSO Indices Based on Patterns of Satellite-Derived Precipitation. J. Climate, 13, 2786–2793, doi: 10.1175/1520-0442(2000)013<2786:EIBOPO>2.0.CO;2

Khodri, M., Izumo, T., Vialard, J., Janicot, S., Cassou, C., Lengaigne, M., Mignot, J., Gastineau, G., Guilyardi, E., Lebas, N., Robock, A. and McPhaden, M. J.: Tropical explosive volcanic eruptions can trigger El Niño by cooling tropical Africa, Nat. Commun., 8(1), 778, doi:10.1038/s41467-017-00755-6, 2017.

Maher, N., McGregor, S., England, M. H. and Gupta, A. S.: Effects of volcanism on tropical variability: EFFECTS OF VOLCANISM, Geophys. Res. Lett., 42(14), 6024–6033, doi:10.1002/2015GL064751, 2015.

Meehl, G. A., Teng, H., Maher, N. and England, M. H.: Effects of the Mount Pinatubo eruption on decadal climate prediction skill of Pacific sea surface temperatures: PINATUBO AND DECADAL PREDICTION SKILL, Geophys. Res. Lett., 42(24), 10,840–10,846, doi:10.1002/2015GL066608, 2015.

Pausata, F. S. R., Chafik, L., Caballero, R. and Battisti, D. S.: Impacts of high-latitude volcanic eruptions on ENSO and AMOC, Proc. Natl. Acad. Sci. U. S. A., 112(45), 13784–13788, doi:10.1073/pnas.1509153112, 2015.

Predybaylo, E., G. L. Stenchikov, A. T. Wittenberg, and F. Zeng: Impacts of a Pinatubo‐Size Volcanic Eruption on ENSO, J. Geophys. Res. Atmos., 122, 925–947, doi:10.1002/2016JD025796, 2017

Swingedouw, D., Mignot, J., Ortega, P., Khodri, M., Menegoz, M., Cassou, C. and Hanquiez, V.: Impact of explosive volcanic eruptions on the main climate variability modes, Glob. Planet. Change, 150, 24–45, doi:10.1016/j.gloplacha.2017.01.006, 2017.

Trenberth, K.E. and D.P. Stepaniak, 2001: Indices of El Niño Evolution. J. Climate, 14, 1697–1701, doi: 10.1175/1520-0442(2001)014<1697:LIOENO>2.0.CO;2

[Figure]

Figure 10: Top row shows the Nino4 index and bottom row shows the ENSO Precipitation Index (ESPI) for the first four prediction years calculated as a 12 month running mean to reduce variance. Left (right) column shows the 2012 (2014) initialized experiments. Error bars show the standard deviation of the ensemble and vertical black lines indicate a significant difference.

**Fig. 1.**

---

## Author Comment (AC2) · 19 Apr 2018

We thank the reviewer for the constructive comments and useful suggestions. Below we answer the different comments of the reviewer. We present all reviewer comments and our answers are given in blue.

**General comments**. This paper investigates how much effect a Pinatubo like eruption would have on decadal climate predictions for a number of variables, and whether the phase of major modes of variability (PDO, NAO) at the time of eruption make a difference to the climate response. The authors find that a volcanic eruption does indeed affect decadal predictions of temperature, precipitation and sea ice, and that some aspects are dependent on the initial state of the climate system, particularly at regional scales. This is an interesting and relevant topic, since the occurrence of a volcanic eruption would likely have a large impact on climate on the timescale of decadal predictions. The authors explain the background and relevance of the research well. I feel that the methods are appropriate to address the question and that the conclusions are supported by the evidence shown. The presentation of figures is generally good.

The article is within the scope of ESD and should be interesting to its readership. I feel that the article is worthy of publication, following the correction of some generally minor issues. My main suggested change relates to presenting ensemble spread to show how consistent results are between ensemble members, which will have implications for how predictable the climate response to eruptions is. Similarly, an indication of the extent to which modes of variability (particularly the NAO and PDO) remain in the same phase or diverge between ensemble members post eruption may shed more light on the differences between the two ensembles. My other suggestions mainly concern clarity in some places and English corrections.

We will address these points in the specific comments below.

**Specific comments**

Abstract - Mention that it is an ensemble of forecasts here.

We added this information in the abstract.
"Each forecast contains an artificial Pinatubo-like eruption starting in June of the first prediction year **and consists of ten ensemble members.**"

Abstract, last line – which simulations have the negative PDO? The 2012 ones? Please make that clearer.

Yes, the 2012 ones. We added 2012 to this sentence.
"In contrast, the forecast initialized **2012** with a negative PDO shows a prolonged cooling in the North Pacific basin."

Abstract: "So the question emerges how predictable is the response of the climate system to future eruptions?" Looking at the ensemble spread would also help address this question- see also comments below. Whilst you may find a volcanic signal in the ensemble mean, the paper does not really give a feel for how consistent these signals are between ensemble members, and this would have implications for how predictable the climate response to volcanic eruptions is.

This is a valid point. We don't address the question of how consistent the signals are between the ensemble members explicitly, but implicitly this is addressed through the applied significance test. All individual ensemble members are input of the t-test and the Pinatubo and Baseline ensembles can only be significantly different if there is some consistency between the individual members.
To get a better feeling of the consistency of the ensemble members are we added ensemble spread information in figures 2, 6, and 8 (see attachment). It confirms, if there is a significant difference between the two experiments, then the ensemble spread is also separated. We feel that the ensemble spread does not add to much new information to the plot, but rather makes them more complicated to read. Therefore, we suggest to keep the old ones and add the plots with ensemble spread to a supplement.

Introduction, first paragraph – is it worth mentioning briefly where the predictability comes from? E.g. is it from certain modes of variability? Or from SSTs? Etc. I assume this list of variables are the things that are able to be predicted rather than the predictors?

We agree that it is worth mentioning where the predictability comes from. We added the following paragraph to the manuscript:
"Decadal predictability comes mainly from the multiyear memory of the ocean. The memory in the ocean arises, for instance, from the persistence of ocean heat content anomalies and from properly initialized ocean dynamics and circulation (e.g. Guemas et al. 2012, Matei et al. 2012). A detailed explanation of the principles behind decadal prediction can be found in Kirtman et al. (2013). "

Pg 2 Line 20, also pg 8 line 20- and the south American monsoon e.g. Joseph and Zeng 2011??, Iles and Hegerl 2013;14

You are right. We added the South American monsoon and the suggested references.

Pg 2 Line 21-22 – is this decade long sea ice response for the large eruptions that occurred around the time of the Little Ice Age, or for twentieth century ones as well?

It is also for the twentieth century ones. Ding et al. (2014) examined CMIP5 historical simulations which cover the time period between 1850 and 2005. During this time five major volcanic eruptions occurred (Krakatau 1883, Santa Maria 1902, Agung 1963, El Chichon 1982, and Pinatubo 1991) and they found a period of increased sea ice after most of them. Gagne et al. (2017) focused on the time period 1960 to 2005 with three large tropical eruptions and came to the same conclusion. In order to make this a bit clearer, we rephrased the sentence as follows.
"Sea ice in the Northern hemisphere is also affected by large volcanic eruptions **that occurred between 1850 and 2005** and SVEs can cause up to a decade of increased Arctic sea ice extent (Ding et al., 2014; Gagné et al., 2017)"

Pg 4 line 32 – What is a "lagged-day method"? Can you explain briefly?

The so called lagged-day method is a procedure to generate an ensemble. To create a spread within the ensemble the initialization date of each individual ensemble member is shifted by one day.
In order to make that clearer we rephrased the sentence:
"For our experiment, we perform two decadal forecasts for ten years with ten ensemble members each. For ensemble generation we use the lagged-day initialization method, which means that the individual ensemble member is started on different start days around the 31st December to spread the ensemble."

Pg 5 first paragraph and throughout – I found the naming of the two experiments some-what confusing (Pinatubo-2012 and Pinatubo-2014), because they are initialised on the

last day of December in 2012 and 2014 respectively, meaning that the eruptions happen in 2013 and 2015 respectively, and that only one day of 2012/ 2014 is actually simulated. I wonder if they would be better off being called Pinatubo 2013 and 2015? Or at least make sure that it is clear that the eruptions happened in summer 2013 and 2015.

We would like to keep the experiment names because they follow the CMIP5 and CMIP6 naming convention for decadal predictions. In order to reduce the chance of possible misunderstandings we added the requested information in experimental setup section.
"One forecast was initialized around December, 31st 2012 (Pinatubo-2012) and the other one around December, 31st 2014 (Pinatubo-2014). **The Pinatubo-like eruption happens in June of the first prediction year, which is June 2013 in case of the Pinatubo-2012 experiment and June 2015 for the Pinatubo-2014 experiment.**"

Figure 1a and pg 4 line 27 – is it possible to plot the observed equivalent of figure 1a if its not too much work?

Attached you find a plot which compares our simulated AOD (left) and the AOD forcing data used in the MPI-ESM simulations contributing to CMIP5 and is based on Sato et al. (1993) and Stechnikov et al. (1998) (Fig. S1). We think this gives also a good impression of how good the aerosol model resembles the observed eruption.
However, we want stretch the point that that our goal was not to simulate the exact Pinatubo eruption, but rather simulate an eruption of a similar size as the Pinatubo eruption (Pinatubo-like). Therefore, we would like to move this plot to a possible supplement.

Figure 1 b and c
- Are the vertical lines showing the initialisation dates in the right place? They should be almost on the dashes for 2013 and 2015 on the x axis given the initialisation happens on the 31st December. - Could you indicate on the plots when the eruptions happen?

The vertical lines were unfortunately misplaced we have now moved them to the right place. We also added hatched lines to indicate where the eruptions happen (see attached Fig. 1).

- Could you explain explicitly why the hatched bars are different from the solid bars before the eruptions? – Is this because the initialized runs diverge from the assimilation run? How different is the phase of the NAO/ PDO between different ensemble members before the eruptions? Can this be shown on the plot?

Yes, after initialization the model runs free and the free running ensemble members diverge from the assimilation run. We added the indices for the individual ensemble members to figure 1. There is a good agreement between the ensemble members for the PDO index before the eruption. The ensemble members for the NAO are much more scattered. This is in agreement to other studies that show that the NAO index is hard to predict even on a seasonal timescale (Buttler et al., 2016)

Also it would be interesting to see how the NAO and PDO indices evolve after the eruptions and how much this differs between ensemble members since this will impact the post volcanic climate response.

We calculated the NAO and PDO indices for the first four prediction years as a twelve month running mean to reduce variance (Fig. S3). The spread of the NAO index is large in all experiments and there is no significant difference between the Baseline1 and Pinatubo simulations. That being said, there is a tendency to a positive NAO index in the Pinatubo-2012 ensemble two to three years after the eruption, which is not visible in the Baseline-2012 ensemble (Fig. S3a). In contrast, the Pinatubo-2014 ensemble shows a tendency to negative a NAO index in this time frame (Fig. S3b). This negative NAO could potentially be the reason for the wetter conditions over Europe in the Pinatubo-2014, but we want to emphasize that these results are not significant. The ensemble spread of the PDO index is also large for all experiments. The spread of the simulations with a Pinatubo-like eruption is bigger than in the Baseline1 simulations in both experiments and the ensemble mean index of the Pinatubo simulations is damped compared to the Baseline1 (Fig. S3 c, d). The ensemble mean of the PDO index diverges stronger in the 2012 initialized ensemble. This could explain the more pronounced horse shoe pattern in near surface temperature in the 2012 experiment. We will add figure S3 to the supplementary material.

Page 6 line 2 "AS it would be issued by the Miklip project" – what do you mean? Is this how the Miklip forecasts are presented? Do you mean that they present things as 4 year means? If not, what is the rationale for presenting results as 4 year means when some aspects of the climate response to volcanoes e.g. the precipitation response over land, happen on a timescale less than 4 years?

Yes, the presenting 4 year means are the standard way in decadal prediction. This is due to the fact that the prediction skill decreases rapidly with shorter temporal smoothing. We understand your concern about the precipitation response over land. Therefore, we prepared Fig. 8 with 2 year means instead of 4 year means also (see attachment). See also the answer to your comment below.

Page 8 : Please define what a "frost day" is. Also, I am not sure whether it is really a climate extreme – i.e. in cold regions every day might be a frost day, and that would be normal, not extreme.

You are right. We removed the term "climate extreme" and added the definition of a frost day. It reads now as follows:
"Not only mean temperature values are influenced by volcanic aerosol, but also the **daily temperature minimum**. The Expert Team of Climate Change Indices (ETCCDI, Karl et al. (1999)) **defines a day as frost day if the daily minimum temperature is below 0°C and the number of frost days (FD) as the sum of those days**. Figure 7 shows the anomaly of the number of frost days in our simulations for the prediction years 1-4."

Page 9 line 8/9: " in most regions" I am not sure that a precipitation reduction can be

seen in "most regions" in figure 9.

We agree. We changed this to "large parts, especially over land, ..." and also added discussion about the ENSO like pattern over Southeast Asia and the East Pacific (see also the answer to your comment below).

Figure2 - "Anomalies are based on the period 1981-2010" – is the climatology based on the assimilation run? - I assume these plots are showing annual means? - It could be interesting to show the ensemble spread as well. (Also for the other similar figures).

We removed the sentence "Anomalies are based on the period 1981-2010" as we are not showing anomalies. The plots are showing 4 year means. We added the ensemble spread to the plots as you suggested (see attachment)

Figure 5: near-zero changes could be shown in white for clarity e.g. so it would be white where there is no sea ice all year round, or sea ice 100

We updated the figure. Near-zero changes are now shown in white.

Figure 9 and related discussion (precipitation):
-Is the difference in the Pacific related to the phase of ENSO?
How does the phase of ENSO compare in the post eruption years between the two ensembles? (I know they started off in the same phase when they were initialised, but they may have deviated since)

Thanks for your comment. We included additional ENSO indices (Nino4 and ESPI, Fig. 10, attached) to the manuscript. Indeed, we find that the Pinatubo-2014 shows a tendency to El Nino conditions in the second prediction year. We therefore added discussion about ENSO to section "3.4 Precipitation". It now reads as follows:

"In the global precipitation maps, we see a decrease of precipitation for both experiments through the volcanic aerosol in large parts, especially over land, in the first four prediction years (Fig. 9). The drying effect is strongest over the tropics, particularly in Southeast Asia, and is even more pronounced in exp-2014. In fact, the tropical precipitation pattern in Southeast Asia and the East Pacific in exp-2014 is very similar to an El Nino response. Recent model studies (Maher et al., 2015; Pausata et al., 2015; Khodri et al., 2017) revealed that volcanic eruptions have a significant impact on ENSO and there is some ongoing debate whether tropical volcanic eruption can trigger an El Nino event (Meehl et al., 2015; Predybaylo et al., 2017; Swingedouw et al., 2017). To further investigate this, we calculated the temperature based Nino4 index (Trenberth and Stepaniak, 2001) and the ENSO precipitation index (ESPI, Curtis and Adler, 2000) for both experiments for the first four prediction years (Fig. 10) as twelve month running means to reduce variance. The ensemble initialized in 2014 with a Pinatubo-like eruption shows a tendency towards El Nino conditions, whereas the baseline1 ensemble favors a weak La Nina condition (Fig. 10 b, d). The difference between the two experiments in the ESPI is significant until simulation months 18-30 when both indices come back to neutral conditions. In exp-2012 there is no difference evident in the first three prediction years, but in year four the baseline1 ensemble starts simulating a La Nina phase (Fig. 10 a, c) with a significant difference to the Pinatubo-like experiment. In general, exp-2014 shows a stronger drying response in the tropical region. In contrast, in this experiment, wetter conditions over Western Europe can be found which does not occur in exp-2012 (Fig. 9)."

-Also, is the difference over Europe due to a different phase of the NAO?

See the answer to your comment above about the evolution of NAO index after the eruption.

-I am also intrigued as to how similar or different the variability of modes like ENSO/ PDO/ NAO are within the members of each ensemble- i.e. the PDO and NAO start off in a similar phase, but how much do they diverge between ensemble members over time?

We calculated ENSO, PDO, and NAO for the individual ensemble members for the first four prediction years. See answers to your comments above.

-I also suspect that using a mean of the four post eruption years is not the most appropriate to look at land precipitation, because the response tends to last more like 2 years, unless presenting 4 year means is the standard for decadal predictions.

Yes, presenting 4 year running means is the standard way to present forecasts in decadal prediction (see also comment above). Nevertheless, we understand your concern about the land precipitation. Therefore, we prepared figure 9 also with a two year mean (see attached Fig S9). The overall pattern looks very similar to the 4 year mean, but has less significant points. As there seem not much added information in the two year means and to keep consistent with the standard we suggest to keep the four year means and add the two year mean version of figure 9 to the supplement.

Pg 10 ENSO discussion: It would be worth mentioning that this influence of volcanoes on ENSO is debated.

Because we added ENSO to section 3.4, we decided to remove this sentence. But we mention there that the influence of volcanoes on ENSO is debated.

Conclusions "We have shown that the global near-surface air temperature and precipitation decrease as a response to the volcanic eruption is independent of the initial conditions" – but only for the initial conditions that you have tested- this sentence currently sounds as if you mean its independent of all initial conditions, including things you haven't tested (e.g. ENSO phase), which you obviously can't comment on.

You are right, this conclusion sounds too strong. We now specifically mention the NAO and PDO in this sentence.
"We have shown that the global near-surface air temperature and precipitation decrease as a response to the volcanic eruption is independent **of the initial state of the PDO and the NAO** and that the reduction is significant for the whole prediction period in both forecasts."

**Technical corrections**

Pg 1 line 9/10 "So the question emerges how predictable is the response of the climate system to future eruptions?" Add a colon between emerges and how. Same for page 3 line 9 -> "the question arises: What would happen."

Changed.

Line 12: "among other things" – if I understand correctly, the only other thing was the phase of the NAO, so this could be said specifically.

Yes, you are right. We changed it and mention the NAO specifically now.

Line 21: "than in the simulations initialized in 2014" – delete "than", otherwise it means the opposite!

Thanks!

Pg2 line6 MetOffice -> Met Office, (assuming you mean the UK one)

Changed to "UK Met Office"

Pg2 line 1 "more and more attention is BEING paid"

Added.

Pg 2 line 16 "regionally, HOWEVER, warm anomalies are ALSO found. . ." ("although" is not the right word and changes the meaning of the sentence). Is it worth saying why? i.e. its related to the positive NAO pattern that often occurs after eruptions.

Thanks for the correction. We decided not to mention the NAO here because it is discussed later in this section.

Pg 2 Line 18 "atmosphere AND ocean dynamics"

Changed.

Pg 2 Line 19 "The last major volcanic eruptions followed a period" -> "The last major volcanic eruptions WERE followed by a period" otherwise it means the opposite. Also,

what do you mean by "the last major eruptions?" The ones that occurred this century? Same for pg 7 line 10

Yes, it refers to the ones in the CMIP5 historical time period, which is from 1850 until 2005. We added the missing information:
"The major volcanic eruptions that occurred since 1850 were followed a period of reduced global precipitation"

Pg 2 Line 25 – are these studies about the NAO observation based or model based?

Ortega et al (2015) and Swingedouw et al. (2017) use reconstruction and reanalysis data, whereas Bittner et al. (2016) uses a large ensemble simulation.

Pg 2 Line 31 "could demonstrate" -> "demonstrated"

Changed

Pg 2 Line 34 "volcanic future eruptions" -> future volcanic eruptions

Changed

Pg 3 line 10 "from the start year" -> "on the start year"

Changed

Pg 3 line 13 "multi-yearly" -> "multi-year"

Changed

Pg 4 line 25 "in the order of 50 to 100 years" -> "in the order of ONCE EVERY 50 to 100 years"

Added

Pg 5 line 7 "are in both experiments in a similar state" -> "are in a similar state in both experiments"

Changed

Pg 5 line 11 variables -> modes

We rephrased the entire sentence.

Pg 5 line 12 "nowadays set up" sounds a bit strange. How about "present day set up"?

We changed it to "present day set up"

Pg 7 line 8 "solar near-IR radiation BY the increased sulfate aerosol"

Added

Pg 7 line 19 (and elsewhere) "reacts stronger" -> "reacts more strongly"

Changed

Pr 8 line 13 "Whereas in the Pinatubo-2014 experiment, the frost days increase most over the whole of North America, Scandinavia, Nordic Sea, and East Asia." I get the impression that this sentence should be joined to the previous one "..., whereas..." If not, the change "whereas" to "In contrast". "Whereas" needs both statements to be in one sentence. Also "Nordic Sea" -> "the Nordic Sea".

Thanks. We joined the sentences and added the missing "the"

Pg 8 line 21 –"...60

We added Lau et al. (2008) as reference

Pg 8 line 30 "area" -> "are a"

Changed.

Pg 8 line 31 "but it turned out that the precipitation changes are..." sounds too colloquial. "but the precipitation changes were..." is sufficient.

We rephrased the sentence as follows.
"Similar behavior has been found in CMIP5 model simulations, although they under-estimated the precipitation changes compared to observational data (Iles et al., 2013;

Iles and Hegerl, 2014; Paik and Min, 2017)."

Pg 10 line 2 – delete "Therefore"

Removed

Pg 10 line 8 "are in both experiments in a similar state" -> are in a similar state in both experiments AT THE TIME OF INITIALISATION. (see also comments for figure 9)

Added.

Pg 10 line 28 "We could not find a clear link between the different initial states of the NAO and one of these changes." "one of these changes" -> "any of these changes" or just "these changes".

Changed.

Pg 11 line 8 "experiments will be conducted, where similar to our experiment, the impact" -> "experiments will be conducted where, similar to our experiment, the impact"

Changed.

Figure 6 and 8 captions "other regions" -> "different regions"

Changed.

**References**

Butler, A. H., Arribas, A., Athanassiadou, M., Baehr, J., Calvo, N., Charlton-Perez, A., et al. (2016). The climate-system historical forecast project: Do stratosphere-resolving models make better seasonal climate predictions in boreal winter? Quarterly Journal of the Royal Meteorological Society, 142(696), 1413–1427.

Curtis, S. and R. Adler, 2000: ENSO Indices Based on Patterns of Satellite-Derived Precipitation. J. Climate, 13, 2786–2793, doi: 10.1175/1520-0442(2000)013<2786:EIBOPO>2.0.CO;2

Ding, Y., Carton, J. A., Chepurin, G. A., Stenchikov, G., Robock, A., Sentman, L. T. and Krasting, J. P.: Ocean response to volcanic eruptions in Coupled Model Intercomparison Project 5 simulations, J. Geophys. Res. C: Oceans, 119(9), 5622–5637, doi:10.1002/2013jc009780, 2014.

Gagné, M.-È., Kirchmeier-Young, M. C., Gillett, N. P. and Fyfe, J. C.: Arctic sea ice response to the eruptions of Agung, El Chichón and Pinatubo: Arctic sea ice response to volcanoes, J. Geophys. Res. D: Atmos., doi:10.1002/2017JD027038, 2017.

Guemas, V., F. J. Doblas-Reyes, F. Lienert, Y. Soufflet, and H. Du, 2012: Identifying the causes of the poor decadal climate prediction skill over the North Pacific. J. Geophys. Res., 117, D20111, doi:https://doi.org/10.1029/2012JD018004.

Iles, C. E. and Hegerl, G. C.: The global precipitation response to volcanic eruptions in the CMIP5 models, Environ. Res. Lett., 9(10), 104012, doi:10.1088/1748-9326/9/10/104012, 2014.

Iles, C. E., Hegerl, G. C., Schurer, A. P. and Zhang, X.: The effect of volcanic eruptions on global precipitation: VOLCANOES AND PRECIPITATION, J. Geophys. Res. D: Atmos., 118(16), 8770–8786, doi:10.1002/jgrd.50678, 2013.

Karl, T. R., Nicholls, N. and Ghazi, A.: CLIVAR/GCOS/WMO Workshop on Indices and Indicators for Climate Extremes Workshop Summary, in Weather and Climate Extremes, pp. 3–7., 1999.

Khodri, M., Izumo, T., Vialard, J., Janicot, S., Cassou, C., Lengaigne, M., Mignot, J., Gastineau, G., Guilyardi, E., Lebas, N., Robock, A. and McPhaden, M. J.: Tropical explosive volcanic eruptions can trigger El Niño by cooling tropical Africa, Nat. Commun., 8(1), 778, doi:10.1038/s41467-017-00755-6, 2017.

Kirtman, B., and Coauthors, 2013: Near-term climate change: Projections and predictability. Climate Change 2013: The Physical Science Basis, T. F. Stocker et al., Eds., Cambridge University Press, 953–1028.
Lau, K., V. Ramanathan, G. Wu, Z. Li, S.C. Tsay, C. Hsu, R. Sikka, B. Holben, D. Lu, G. Tartari, M. Chin, P. Koudelova, H. Chen, Y. Ma, J. Huang, K. Taniguchi, and R. Zhang: The Joint Aerosol–Monsoon Experiment: A New Challenge for Monsoon Climate Research. Bull. Amer. Meteor. Soc., 89, 369–384, doi: 10.1175/BAMS-89-3-369, 2008

Maher, N., McGregor, S., England, M. H. and Gupta, A. S.: Effects of volcanism on tropical variability: EFFECTS OF VOLCANISM, Geophys. Res. Lett., 42(14), 6024–6033, doi:10.1002/2015GL064751, 2015.

Matei, D., J. Baehr, J. H. Jungclaus, H. Haak, W. A. Müller, and J. Marotzke, 201a: Multiyear prediction of monthly mean Atlantic meridional overturning circulation at 26.5°N. Science, 335, 76–79, doi: 10.1126/science.1210299.

Meehl, G. A., Teng, H., Maher, N. and England, M. H.: Effects of the Mount Pinatubo eruption on decadal climate prediction skill of Pacific sea surface temperatures, Geophys. Res. Lett., 42(24), 10,840–10,846, doi:10.1002/2015GL066608, 2015.

Paik, S. and Min, S.-K.: Climate responses to volcanic eruptions assessed from observations and CMIP5 multi-models, Clim. Dyn., 48(3-4), 1017–1030, doi:10.1007/s00382-016-3125-4, 2017.

Pausata, F. S. R., Chafik, L., Caballero, R. and Battisti, D. S.: Impacts of high-latitude volcanic eruptions on ENSO and AMOC, Proc. Natl. Acad. Sci. U. S. A., 112(45), 13784–13788, doi:10.1073/pnas.1509153112, 2015.

Predybaylo, E., G. L. Stenchikov, A. T. Wittenberg, and F. Zeng: Impacts of a Pinatubo‐Size Volcanic Eruption on ENSO, J. Geophys. Res. Atmos., 122, 925–947, doi:10.1002/2016JD025796, 2017

Sato M., J. Hansen, M. McCormick, and J. Pollack, Stratospheric aerosol optical depths, J. Geophys. Res., 98, 22,987–22,994, doi:10.1029/93JD02553, 1993

Stenchikov, G. L., I. Kirchner, A. Robock, H.-F. Graf, J. C. Antuña, R. G. Grainger, A.
Lambert, and L. Thomason, Radiative forcing from the 1991 Mount Pinatubo volcanic eruption, J. Geophys. Res., 103, 13,837–13,858, doi:10.1029/98JD00693, 1998

Swingedouw, D., Mignot, J., Ortega, P., Khodri, M., Menegoz, M., Cassou, C. and Hanquiez, V.: Impact of explosive volcanic eruptions on the main climate variability modes, Glob. Planet. Change, 150, 24–45, doi:10.1016/j.gloplacha.2017.01.006, 2017.

Trenberth, K.E. and D.P. Stepaniak, 2001: Indices of El Niño Evolution. J. Climate, 14, 1697–1701, doi: 10.1175/1520-0442(2001)014<1697:LIOENO>2.0.CO;2

[Figure]

*Figure 1: a) Temporal evolution of the stratospheric aerosol optical depth (AOD) used for our simulations with a Pinatubo-like eruption. b), and c) show the climate indices NAO (a) and PDO (b) calculated from the assimilation model run. Hatched and lighter colors are the indices of the prediction system ensemble mean before the Pinatubo-like eruption and grey circles are the individual ensemble members. Vertical lines indicate the initialization time of our forecast experiments and vertical hatched lines indicate the start of the Pinatubo-like eruption.*
*This figure will replace Fig. 1 in the manuscript.*

**Fig. 1.**

[Figure]

[Figure]

*Figure 10: Top row shows the Nino4 index and bottom row shows the ENSO Precipitation Index (ESPI) for the first four prediction years calculated as a 12 month running mean to reduce variance. Left (right) column shows the 2012 (2014) initialized experiments. Error bars show the standard deviation of the ensemble and vertical black lines indicate a significant difference.*
*This figure will be added as Fig. 10 to the manuscript.*

**Fig. 2.**

[Figure]

*Figure S1: Left: Simulated AOD as in Fig. 1a), Right: AOD forcing data used in the MPI-ESM simulations contributing to CMIP5 and is based on Sato et al. (1993) and Stenchikov et al. (1998).*
*This figure will be added to the supplementary material.*

**Fig. 3.**

[Figure]

*Figure S2: Like Fig. 2, but with standard deviation.*
*This figure will be added to the supplementary material.*

**Fig. 4.**

[Figure]

*Figure S3: Top row shows the NAO index and bottom row shows the PDO index for the first four prediction years calculated as a 12 month running mean to reduce variance. Left (right) column shows the 2012 (2014) initialized experiments. Error bars show the standard deviation of the ensemble and vertical black lines indicate a significant difference.*
*This figure will be added to the supplementary material.*

**Fig. 5.**

[Figure]

[Figure]

*Figure S6: Like Fig. 6 in the manuscript, but with standard deviation.*
*This figure will be added to the supplementary material.*

**Fig. 6.**

[Figure]

[Figure]

Figure S8: Like Fig. 8 in the manuscript, but with standard deviation.
This figure will be added to the supplementary material.

**Fig. 7.**

[Figure]

*Figure S8: Like Fig. 8 in the manuscript, but for 2-year running means, instead of 4-year means.*
*This figure will be added to the supplementary material.*

**Fig. 8.**

**(a)** Initialized 2012     **(b)** Initialized 2014     **(c)**     2012 - 2014

[Figure]

[Figure]

[Figure]

*Figure S9: Like Fig. 9 in the manuscript, but for prediction year 1-2 instead of 1-4.*
*This figure will be added to the supplementary material.*

**Fig. 9.**

---

## Author Comment (AC3) · 19 Apr 2018

We thank the reviewer for the constructive comments and useful suggestions. Below we answer the different comments of the reviewer. We present all reviewer comments and our answers are given in blue.

**General Comments:**

This manuscript describes a set of decadal prediction experiments initialized with different phases of the NAO PDO where the effects of a volcanic eruption of the magnitude of Mt. Pinatubo are examined. The authors find expected responses in the temperature and precipitation fields in the global average, but find that the different initial states

produce regional responses that are different. This is a nice result that has implications for how decadal predictions should be performed when a volcanic eruption happens.

The manuscript can be improved in terms of clarity and improvements in figures.

The paper is more or less complete in terms of analysis except for one issue that they could have explored a little more. This pertains to the 2014-Pinatubo experiment where the years 1-4 precipitation response (Figure 9) indicates a pattern that is very similar to an El Niño response. While this hints at a possible volcano triggered El Niño, they make no mention of it but discuss this possibility in the conclusions purely in other published references. It would be interesting to see if indeed the 2014-Pinatubo runs show that an ENSO event was triggered – while the 2012-Pinatubo doesn't.

We will address these points in the specific comments below.

**Detailed comments:**

1. Abstract. Line 12: A little more descriptive wording than just "the MiKlip prediction system" would be useful here.

We added a short explanation. A more detailed explanation of the MiKlip prediction system can be found in section 2.1.

2. Page 2, Line 1: Suggest changing "more attention is paid to the research field of decadal climate..." to "more attention is paid to decadal climate..."

We have accepted the proposed change.

3. Page 2, Line 18: The wording in "impact on atmospheric composition, atmosphere, ocean dynamics" does not make clear what other than composition is altered in the atmosphere.

You are right. There should be an AND instead of the comma. We changed it to: "SVEs also have an impact on atmospheric composition, atmosphere **and** ocean dynamics and on the hydrological cycle"

4. Page 2, Line 24: Not clear what a "positive impact on the North Atlantic Oscillation" means. The following sentence does not state it any better when one reads "...the positive NAO response could be better interpreted in terms of a deficit of negative NAO circulations".

We changed the first sentence to make it more clear and removed the second one:
"There is some evidence **from observations and reconstructions that SVEs lead to a positive phase of the Northern Atlantic Oscillation (NAO)** in the first few winters following the eruption (e.g. Ortega et al., 2015; Swingedouw et al., 2017), but recent model studies suggest that this signal might not be that robust (Bittner et al., 2016; Ménégoz et al., in press)."

5. Page 3, Line 9: The use of "nowadays" sounds strange. Suggest rewording.

We reworded "nowadays" to "these days"
"..., the question arises what would happen if a large volcanic eruption occurs these days..."

6. Page 3, Lines 9-10: The sentence "...how dependent are the results from the start year and respectively the initial climate state?" implies that initial climate state and start year are delinked.

You are right. We rephrased the sentence.
"What would happen if a large volcanic eruption occurs these days and how dependent are the results on the start year and the associated initial climate state?"

7. Page 4, Line 3: The phrase "Pinatubo forecasts" should be changed or an uninformed reader may take it as a forecast of the eruption itself.

To avoid misunderstandings we rephrased the sentence.
"We perform our forecasts containing a Pinatubo-like eruption with the ..."

8. Page 4, Line 11: "For decadal forecasting, a stand-by model system for rapid modelbased assessment..." Why the use of "stand-by"? Do you mean "operational"?

We agree, "operational" is the better wording.

9. Page 4, Lines 24-25: Suggest changing "...of the historical time period (1850 till today)" to "...since 1850".

We changed it as you suggested.

10. Page 4, Line 32: "...using a lagged-day method" can be explained better in words or needs a reference.

The so called lagged-day method is a method to generate an ensemble. To create a spread in the ensemble the initialization date of each individual ensemble member is shifted by one day.
In order to make that clearer we rephrased the sentence:
"For our experiment, we perform two decadal forecasts for ten years with ten ensemble members each. For ensemble generation we use the lagged-day initialization method, which means that the individual ensemble member is started on different start days around the 31st December to spread the ensemble."

11. Page 5, Line 1: Why "around"? Shouldn't it be "on"? How many days before and after Dec 31 are the other forecasts?

See above comment.

12. Page 5, Line 11: What does "both variables" refer to?

"Both variables" refers to the previous mentioned indices NAO and PDO. To make that clearer we rephrased it a bit.
"The different phases of the NAO and PDO at initialization time enables us to investigate..."

13. Page 5, Line 12: What is a "a 'nowadays' setup"?

With nowadays setup we meant that we start the model with present day greenhouse gas conditions. We changed "nowadays set up" to "present day set up"

14. Page 6, Lines 5, 6: The word "significant" here (at least in the first instance) needs to be changed to "statistically significant at x

We changed it as you suggested.

15. Page 6, Line 29: Does the phrase "in less warm air being advected…" mean less advection or the air is less warm?

It means "less advection". To make that more clear we changed the sentence as follows:
"The negative PDO phase results **in a reduced advection, which means less** warm air being advected from the North Pacific into this region, …"

16. Page 7, Line 27: Suggest using "On the other hand" instead of "On the contrary".

Changed.

17. Page 8, Line 5: Not sure the difference between what and what stays nearly constant?

We talk about the difference between the Pinatubo and Baseline experiment. For instance, in Fig. 6b and c we find increased values of sea ice in the Pinatubo experiments and the difference between the two stays nearly constant for the whole simulation time. We changed the sentence as follows:
"That is, if there are increased values of SIC in one experiment (Fig. 6b, c, f), the difference of SIC values between the Pinatubo and the Baseline1 simulations stays nearly constant for all prediction years."

18. Page 8, Line 24: Not clear what variable/difference "…stays significant for all lead-times."

We added the missing information to the sentence.
"The **effect of reduced global mean precipitation due to the Pinatubo-like eruption** decreases with prediction time, but stays significant for all lead-times."

[Figure]

19. Page 8, Line 31: Suggest rewording "Similar behavior has been found in CMIP5 model simulations, but it turned out that the precipitation..." to "Similar behavior has been found in CMIP5 model simulations, although they underestimate the precipitation...".

We accepted your suggestion and changed the sentence as follows.
"Similar behavior has been found in CMIP5 model simulations, **although they underestimated the precipitation changes** compared to observational data (Iles et al., 2013; Iles and Hegerl, 2014; Paik and Min, 2017)."

20. Page 9, Lines 9-10: "The drying effect is strongest over the tropics, particularly in Southeast Asia, and is even more pronounced in exp-2014. In general, exp-2014 shows a stronger drying response in the tropical region." I am not sure it is a drying all across the tropics. I see the pattern in exp-2014 as an eastward shift of precipitation - a possible signature of El Niño. The interpretation of Fig. 9 presented needs to a relook and the authors can examine whether indeed this is an ENSO event in the model simulations.

Thanks for your comment. We included additional ENSO indices (Nino4 and ESPI, Fig. 10, attached) to the manuscript. Indeed, we find that the Pinatubo-2014 shows a tendency to El Nino conditions in the second prediction year. We therefore added discussion about ENSO to section "3.4 Precipitation". It now reads as follows:

"In the global precipitation maps, we see a decrease of precipitation for both experiments through the volcanic aerosol in large parts, especially over land, in the first four prediction years (Fig. 9). The drying effect is strongest over the tropics, particularly in Southeast Asia, and is even more pronounced in exp-2014. In fact, the tropical precipitation pattern in Southeast Asia and the East Pacific in exp-2014 is very similar to an El Nino response. Recent model studies (Maher et al., 2015; Pausata et al., 2015; Khodri et al., 2017) revealed that volcanic eruptions have a significant impact on ENSO and there is some ongoing debate whether tropical volcanic eruption can trigger an El

Nino event (Meehl et al., 2015; Predybaylo et al., 2017; Swingedouw et al., 2017). To further investigate this, we calculated the temperature based Nino4 index (Trenberth and Stepaniak, 2001) and the ENSO precipitation index (ESPI, Curtis and Adler, 2000) for both experiments for the first four prediction years (Fig. 10) as twelve month running means to reduce variance. The ensemble initialized in 2014 with a Pinatubo-like eruption shows a tendency towards El Nino conditions, whereas the baseline1 ensemble favors a weak La Nina condition (Fig. 10 b, d). The difference between the two experiments in the ESPI is significant until simulation months 18-30 when both indices come back to neutral conditions. In exp-2012 there is no difference evident in the first three prediction years, but in year four the baseline1 ensemble starts simulating a La Nina phase (Fig. 10 a, c) with a significant difference to the Pinatubo-like experiment. In general, exp-2014 shows a stronger drying response in the tropical region. In contrast, in this experiment, wetter conditions over Western Europe can be found which does not occur in exp-2012 (Fig. 9)."

21. Page 10, Line 8: The phrase "...are in both experiments in a similar state..." may be reworded as "are in a similar state in both experiments".

Changed.

22. Page 10, Lines 31-32: "Therefore our simulations in this study should be extended with experiments starting with other initial conditions like the recent El Nino year 2015/2016." In your simulations, the ENSO response like pattern is seen for 2014-Pinatubo runs. The initial conditions for both runs were similar in terms of ENSO state - yet one of them produces what looks like an ENSO-like pattern. It should be relatively easy to check (and at least comment on) whether it triggered an El Nino. Initializing with ENSO conditions is not answering the question whether volcanoes might trigger ENSO events.

You are right. As mentioned above we included ENSO indices I section 3.4 and included discussion about volcanic triggered ENSO events.

Therefore, we rephrased the sentence in the conclusions:
"We only investigated the volcanic response to different initial conditions of the NAO and PDO. Therefore, our simulations in this study should be extended with experiments starting with other initial conditions like the recent El Nino year 2015/2016."

23. Page 24, Figure 7 caption: What does "two other" refer to?

There was a "the" missing. "The two" refers to the figures on the left and in the middle. "Differences of ensemble mean forecasts of frost days (FD) for prediction years 1-4. Left shows exp-2012 (Pinatubo-2012 - b1-2012), middle shows exp-2014 (Pinatubo-2014 - b1-2014), and right the difference **between the two** (exp-2012 - exp-2014)."

**Technical Comments:**

24. Page 19, Fig 2 caption: The coordinates "North Atlantic (60W, 0E, 50N, 65N)" is better written as "North Atlantic (60W-0E, 50N-65N)". Similarly for other regions.

Changed.

25. Figures 3,5, 7, 9 all have maps shown with no latitude/longitude markings or labels.

Yes, we left them out on purpose. As we show coast lines and in most cases maps of the whole globe, we think adding latitude/longitude labels would redundant information.

26. Figure 4: Vertical axes in hPa units would be better. Latitude axis can be shown with ticks/labels so effects in polar/tropical regions are better seen.

We changed the unit and added more x-ticks.

**References:**

Bittner, M., Schmidt, H., Timmreck, C. and Sienz, F.: Using a large ensemble of simulations to assess the Northern Hemisphere stratospheric dynamical response to tropical volcanic eruptions and its uncertainty, Geophys. Res. Lett., 43(17), 9324–9332, doi:10.1002/2016GL070587, 2016.

Curtis, S. and R. Adler, 2000: ENSO Indices Based on Patterns of Satellite-Derived Precipitation. J. Climate, 13, 2786–2793, doi: 10.1175/1520-0442(2000)013<2786:EIBOPO>2.0.CO;2

Khodri, M., Izumo, T., Vialard, J., Janicot, S., Cassou, C., Lengaigne, M., Mignot, J., Gastineau, G., Guilyardi, E., Lebas, N., Robock, A. and McPhaden, M. J.: Tropical explosive volcanic eruptions can trigger El Niño by cooling tropical Africa, Nat. Commun., 8(1), 778, doi:10.1038/s41467-017-00755-6, 2017.

Iles, C. E. and Hegerl, G. C.: The global precipitation response to volcanic eruptions in the CMIP5 models, Environ. Res. Lett., 9(10), 104012, doi:10.1088/1748-9326/9/10/104012, 2014.

Iles, C. E., Hegerl, G. C., Schurer, A. P. and Zhang, X.: The effect of volcanic eruptions on global precipitation: VOLCANOES AND PRECIPITATION, J. Geophys. Res. D: Atmos., 118(16), 8770–8786, doi:10.1002/jgrd.50678, 2013.

Maher, N., McGregor, S., England, M. H. and Gupta, A. S.: Effects of volcanism on tropical variability: EFFECTS OF VOLCANISM, Geophys. Res. Lett., 42(14), 6024–6033, doi:10.1002/2015GL064751, 2015.

Meehl, G. A., Teng, H., Maher, N. and England, M. H.: Effects of the Mount Pinatubo eruption on decadal climate prediction skill of Pacific sea surface temperatures, Geophys. Res. Lett., 42(24), 10,840–10,846, doi:10.1002/2015GL066608, 2015.

Ménégoz, M., Cassou, C., Swingedouw, D., Bretonnière, P.-A. and Doblas-Reyes, F.: Role of the Atlantic Multidecadal Variability in modulating the climate response to a
Pinatubo-like volcanic eruption, Clim. Dyn., in press

Ortega, P., Lehner, F., Swingedouw, D., Masson-Delmotte, V., Raible, C. C., Casado, M. and Yiou, P.: A model-tested North Atlantic Oscillation reconstruction for the past millennium, Nature, 523(7558), 71–74, doi:10.1038/nature14518, 2015.

Paik, S. and Min, S.-K.: Climate responses to volcanic eruptions assessed from observations and CMIP5 multi-models, Clim. Dyn., 48(3-4), 1017–1030, doi:10.1007/s00382-016-3125-4, 2017.

Pausata, F. S. R., Chafik, L., Caballero, R. and Battisti, D. S.: Impacts of high-latitude volcanic eruptions on ENSO and AMOC, Proc. Natl. Acad. Sci. U. S. A., 112(45), 13784–13788, doi:10.1073/pnas.1509153112, 2015.

Predybaylo, E., G. L. Stenchikov, A. T. Wittenberg, and F. Zeng: Impacts of a Pinatubo–Size Volcanic Eruption on ENSO, J. Geophys. Res. Atmos., 122, 925–947, doi:10.1002/2016JD025796, 2017

Swingedouw, D., Mignot, J., Ortega, P., Khodri, M., Menegoz, M., Cassou, C. and Hanquiez, V.: Impact of explosive volcanic eruptions on the main climate variability modes, Glob. Planet. Change, 150, 24–45, doi:10.1016/j.gloplacha.2017.01.006, 2017.

Trenberth, K.E. and D.P. Stepaniak, 2001: Indices of El Niño Evolution. J. Climate, 14, 1697–1701, doi: 10.1175/1520-0442(2001)014<1697:LIOENO>2.0.CO;2

[Figure]

[Figure]

Figure 10: Top row shows the Nino4 index and bottom row shows the ENSO Precipitation Index (ESPI) for the first four prediction years calculated as a 12 month running mean to reduce variance. Left (right) column shows the 2012 (2014) initialized experiments. Error bars show the standard deviation of the ensemble and vertical black lines indicate a significant difference.

**Fig. 1.**